# Screening of *Helicoverpa armigera* Mobilome Revealed Transposable Element Insertions in Insecticide Resistance Genes

**DOI:** 10.3390/insects11120879

**Published:** 2020-12-11

**Authors:** Khouloud KLAI, Benoît CHÉNAIS, Marwa ZIDI, Salma DJEBBI, Aurore CARUSO, Françoise DENIS, Johann CONFAIS, Myriam BADAWI, Nathalie CASSE, Maha MEZGHANI KHEMAKHEM

**Affiliations:** 1Laboratory of Biochemistry and Biotechnology (LR01ES05), Faculty of Sciences of Tunis, University of Tunis El Manar, Tunis 1068, Tunisia; khouloud.klai@fst.utm.tn (K.K.); marwa.zidi@fst.utm.tn (M.Z.); salma.djebbi@fst.utm.tn (S.D.); 2EA2160 Mer Molécules Santé, Le Mans Université, 72085 Le Mans, France; Benoit.Chenais@univ-lemans.fr (B.C.); Aurore.Caruso@univ-lemans.fr (A.C.); fdenis@univ-lemans.fr (F.D.); Myriam.Badawi@univ-lemans.fr (M.B.); 3URGI, INRAE, Université Paris-Saclay, 78026 Versailles, France; johann.confais@inrae.fr; 4Plant Bioinformatics Facility, BioinfOmics, INRAE, Université Paris-Saclay, 78026 Versailles, France

**Keywords:** *Helicoverpa armigera*, transposable elements, insertions sites, insecticide resistance genes

## Abstract

**Simple Summary:**

Transposable elements (TEs) are mobile DNA sequences that can copy themselves within a host genome. TE-mediated changes in regulation can lead to massive and rapid changes in expression, responses that are potentially highly adaptive when an organism is faced with a mortality agent in the environment, such as an insecticide. *Helicoverpa armigera* shows a hight number of reported cases of insecticide resistance worldwide, having evolved resistance against pyrethroids, organophosphates, carbamates, organochlorines, and recently to macrocyclic lactone spinosad and several *Bacillus thuringiensis* toxins. In the present study, we conducted a TE annotation using combined approaches, and the results revealed a total of 8521 TEs, representing 236,132 copies, covering 12.86% of the *H. armigera* genome. In addition, we underlined TE insertions in defensome genes and we successfully identified nine TE insertions belonging to the RTE, R2, CACTA, Mariner and hAT superfamilies.

**Abstract:**

The cotton bollworm *Helicoverpa armigera* Hübner (*Lepidoptera: Noctuidae*) is an important pest of many crops that has developed resistance to almost all groups of insecticides used for its management. Insecticide resistance was often related to Transposable Element (TE) insertions near specific genes. In the present study, we deeply retrieve and annotate TEs in the *H. armigera* genome using the Pipeline to Retrieve and Annotate Transposable Elements, PiRATE. The results have shown that the TE library consists of 8521 sequences representing 236,132 TE copies, including 3133 Full-Length Copies (FLC), covering 12.86% of the *H. armigera* genome. These TEs were classified as 46.71% Class I and 53.29% Class II elements. Among Class I elements, Short and Long Interspersed Nuclear Elements (SINEs and LINEs) are the main families, representing 21.13% and 19.49% of the total TEs, respectively. Long Terminal Repeat (LTR) and Dictyostelium transposable element (DIRS) are less represented, with 5.55% and 0.53%, respectively. Class II elements are mainly Miniature Inverted Transposable Elements (MITEs) (49.11%), then Terminal Inverted Repeats (TIRs) (4.09%). Superfamilies of Class II elements, i.e., Transib, P elements, CACTA, Mutator, PIF-harbinger, Helitron, Maverick, Crypton and Merlin, were less represented, accounting for only 1.96% of total TEs. In addition, we highlighted TE insertions in insecticide resistance genes and we successfully identified nine TE insertions belonging to RTE, R2, CACTA, Mariner and hAT superfamilies. These insertions are hosted in genes encoding cytochrome P450 (CyP450), glutathione S-transferase (GST), and ATP-binding cassette (ABC) transporter belonging to the G and C1 family members. These insertions could therefore be involved in insecticide resistance observed in this pest.

## 1. Introduction

The cotton bollworm, *Helicoverpa armigera* (Hübner) (*Lepidoptera, Noctuidae*), is a serious crop pest having a worldwide distribution [1]. This polyphagous insect causes substantial damages to a wide range of hosts, including cotton, maize, sorghum, and tomato [2]. The biological and ecological traits of *H. armigera*, such as high reproduction rate, polyphagy, high mobility and facultative diapause, make it difficult to control [3]. Management of *H. armigera* attacks rely heavily on the use of chemicals [4]. However, this practice is harmful to the environment and has caused a rapid buildup of insecticide resistance in *H. armigera* populations [5].

Insecticide resistance in *H. armigera* is widespread and has evolved against most of commonly used insecticides [6]. To survive, pests have developed various mechanisms to resist against toxic compounds. These mechanisms include point mutations resulting in target-site resistance such as knockdown resistance (kdr), acetylcholinesterase (Ace-1) and receptor sub-unit termed (RDL) mutations, and also metabolic resistance with involvement of several detoxification enzymes [7].

Metabolic detoxification of toxins is the primary strategy occurring in three phases, each with its own set of enzymes or transporters. Cytochrome P450 monooxygenases (P450s) and carboxylesterases (CarE) carry out phase I, glutathione S-transferases (GSTs) and UDP-glycosyltransferases (UGTs) are phase II enzymes, and ATP-binding cassette transporters (ABC) ensure phase III [8,9,10]. The understanding of resistance mechanisms remains a challenge that next generation sequencing technologies and the increasing number of sequenced genomes can help to address [11].

Transposable Elements (TEs) are ubiquitous components of eukaryotic genomes that are strongly regulated and inactivated by mutations, which keep transposition events relatively rare [12,13]. However, because of their ability to replicate, TEs may accumulate in host genomes and generate abundant sites for chromosomal rearrangements, which may have deleterious or beneficial consequences [14]. In addition, TEs can provide a selective advantage through their insertion sites, which can enhance or repress gene expression or can be domesticated as new host gene [14,15,16]. Thus, TEs are an important source of variability for the genomes of their hosts and are therefore key to understanding their evolution. Indeed, TEs may be involved in the genetic adaptation of organisms such as insects to stressful environments, among which is the acquisition of insecticide resistance [17].

Several studies have shown that insecticide resistance can be associated with TE insertions in specific genes. For example, dichlorodiphenyltrichloroethane (DTT) resistance in *Drosophila melanogaster* was correlated with the insertion of a Long Terminal Repeat (LTR)-gypsy retroelement into the 5′ region of the cytochrome P450 gene [18,19]. In *Helicoverpa zea,* several TE insertions in regulatory regions, exons and introns of cytochrome P450 genes were related with pyrethroid resistance and xenobiotic metabolism [20].

TEs are classified into two major classes depending on the transposition intermediate. Class I, or retroelements, replicate and transpose via an RNA intermediate; while Class II elements, or DNA transposons, are mobilized via a DNA intermediate [21]. According to the classification of Wicker et al. [22], each class is subdivided into orders and superfamilies. Class I elements are further subdivided into long terminal repeat (LTR) retrotransposons and non-LTR retrotransposons. The non-LTR retrotransposons include the long interspersed nuclear elements (LINEs) and the short interspersed nuclear elements (SINEs) as well as the Penelope-Like Elements (PLEs). Class II elements are subdivided into two subclasses. Subclass 1 includes the terminal inverted repeat (TIR) transposons and Crypton-like elements, which cleave both DNA intermediate strands, while subclass 2 elements including Mavericks and Helitrons with a single-strand DNA intermediate have a replicative mode of transposition [22,23,24]. Class II TIRs transposons also include Miniature Inverted-repeat Transposable Elements (MITEs), which are short (~100 to 800 bp) non-autonomous truncated versions of autonomous transposable elements. MITEs possess conserved terminal inverted repeats (TIRs ≥ 10 bp) and a target site duplication (TSDs = 2~10 bp) [25,26].

Annotation of TEs is a challenging task because of their diversity, their repetitive nature and the complexity of their structures, and numerous tools have been designed to identify TEs [27]. In this study, we used the Pipeline to Retrieve and Annotate Transposable Elements (PiRATE) [28] to annotate the mobilome of *H. armigera* and pinpoint TEs inserted in defensome genes.

## 2. Material and Methods

### 2.1. Mobilome Annotation

The *H. armigera* genome available in GenBank-NCBI (BioProject PRJNA378437) is 337,087 Mb. This genome is assembled in 24,552 contigs and 998 scaffolds corresponding to 23.5 kb and 1000 kb N50 length respectively [29].

*H. armigera* genome assembly and the corresponding raw Illumina data were both submitted to the PiRATE pipeline to search for TEs following three steps [28]. In the first step, putative TE sequences were detected using four approaches. The first approach is a similarity-based detection of TEs using RepeatMasker [30] and TE-HMMER [31]. The second approach is a structure-based detection, using LTRharvest [32], MGEScan-nonLTR [33] and SINE-Finder [34]. The repetitiveness-based detection is the third approach using TE *de novo* [35] and Repeat Scout [36]. The last approach is a *de novo* approach using the dnaPipeTE tool [37].

After TE detection, a second step was performed to eliminate redundant sequences and classify the remaining sequences using the PASTEC tool following the Wicker’s 80-80-80 rules corresponding to sequences longer than 80 bp, sharing more than 80% sequence identity and over 80% of their length [22,38]. Two libraries were generated: a “total TEs library” containing the potentially autonomous TEs and the non-autonomous TEs, and a “repeated elements library” containing the uncategorized repeated sequences and the non-TE sequences. Subsequently, two runs of TEannot [35] were performed for each library to generate the final libraries of total TEs and total repeats. To refine the annotation of TE copies in the whole genome, we used the TEannot pipeline from the REPET package v3.0 with TEs sequences of PiRATE step 1 that align at least with one Full-Length Copy (FLC) on the genome assembly [35,39,40].

Finally, a manual curation was released for all annotated TEs to find corresponding families. This analysis was performed by nucleotide Basic Local Alignment Search Tool (BLAST) against Repbase (48,225 TE sequences) and Dfam (6959 sequences) databases using a threshold value of 80% (Figure 1).

To identify putative MITE sequences, the *H. armigera* genome assembly was submitted to the MITE Tracker tool [41] (Figure 1). This tool searches for putative inverted repeat sequences ranging from 50 to 800 bp. Subsequently, putative MITEs were aligned and clustered into families by Vsearch [42] based on target sites duplication (TSD) and Terminal Inverted Repeat (TIR) sequences.

### 2.2. Search for TE Insertions in Defensome Genes

Annotated TEs from the *H. armigera* genome have been extended by 50 kb both upstream and downstream of their DNA sequences. The nucleotide BLAST was used to find defensome genes in the extended regions using 80% similarity and 80% query coverage threshold (Figure 1).

## 3. Results

### 3.1. TEs Annotation of the H. armigera Genome

The screening of TEs in the *H. armigera* genome using different detection tools led to the identification of 100,184 TE candidates after redundance elimination. The classification step generated two libraries: the “total TE library” and the “total repeats library” containing 4336 sequences (11.36%) and 5201 sequences (13.63%), respectively. Among the TE library, 3133 sequences were identified as FLC belonging to Class I (2720 FLC), Class II (413 FLC) and 61 sequences were undefined (Appendix A). A total of 70,030 sequences was classified as non-TE (41,411 sequences) and unclassified TEs (28,619 sequences). The results revealed that TE sequences cover 12.86% (43,349,853 bp) of the *H. armigera* genome and most of the TE sequences belong to Class II elements, accounting for 53.29% of the total TE content, while Class I elements account for 46.71%.

Among Class I elements, SINEs and LINEs were the main families, representing 21.13% and 19.49% of the total TEs, respectively. LTR elements were represented with 5.55% and Dictyostelium transposable element (DIRS) with only 0.53%. The Class II elements were represented mainly by MITEs and TIRs with 49.11% and 4.09%, respectively.

To investigate the evolutionary history of TEs in the *H. armigera* genome, we plotted the distribution of identity values between copies and their representative sequences. Distributions of TE classes showed a peak at 80% identity for Class I elements, while, for Class II elements, the distribution was linear with a recent burst at 98% identity (Figure 2).

This analysis revealed that the *H. armigera* genome has undergone a multitude of ancient and recent bursts of different TE superfamilies showing its fluidity. Distributions of TE copies showed three peaks of transposition activity (Figure 3). The first peak is at 65% divergence involving a burst of the DIRS order. We also noted a second TE burst, particularly for LINE, SINE, LTR, TIR, and Helitron orders at 80% identity. In addition, the distribution also showed the appearance of MITE elements at 95% identity, suggesting a recent invasion of the *H. armigera* genome by these TEs (Figure 3).

#### 3.1.1. Class I Retrotransposons

The annotation of Class I retrotransposons in the *H. armigera* genome allowed for the identification of 3980 sequences representing 186,645 copies belonging to 10 superfamilies (Table 1).

From the LTR retrotransposons, 345 TEs were full-length copies (FLC). Gypsy was the most abundant LTR superfamily with 241 sequences and 14,876 copies followed by Bel-Pao (155 sequences) and Copia (77 sequences) with 97 and 67 FLC, respectively. BLAST searches against Repbase and Dfam databases showed that, for all identified LTR sequences, no similarity was found with TEs in databases.

Regarding LINEs retrotransposons, the most abundant elements belong to the Jockey and RTE superfamilies, with 716 and 604 sequences, respectively, representing 44,432 and 46,458 copies, respectively. The RTE superfamily contains the highest number of FLCs among all annotated TEs (Table 1). According to the BLAST searches, no similarity was found for Jockey elements in databases while 17 RTE sequences showed similarity ranging from 86% to 100% with RTE-1_Avan, RTE-2_Hmel_C, RTE-3_DPl, RTE-4_DPl, RTE-5_DPl and Proto2-1_BM families (Appendix A).

Concerning SINEs retrotransposons, the tRNA-derived SINE superfamily corresponds to 45.15% of all Class I TEs with 1797 sequences representing 51,581 copies and 1120 FLCs. Among these sequences, 578 SINEs belong to the HaSE1 family while 85 sequences fell into the HaSE3 family with a similarity ranging from 80% to 100% (Appendix A).

#### 3.1.2. Class II Transposons

Our results revealed that DNA transposons in the *H. armigera* genome are represented by a total of 4541 sequences representing 49,487 copies belonging to 13 superfamilies (Table 1).

Regarding TIR elements, the hAT superfamily was the most abundant, including 120 sequences, among which 105 elements were FLC. BLAST searches against Repbase and Dfam databases revealed high similarity of three hAT elements with the hAT-1_DAN family (Appendix A).

The Tc1/mariner superfamily was represented by 106 elements (2284 copies), among which 74 FLC. Research using BLAST has shown that 12 of these elements are distributed among five families—ANM4, nMar-2_Avan, nMar-18_Hmel, Mariner-1_AMel and Mariner-3_BM—exhibiting similarity ranging from 80.25% to 95.34% (Appendix A).

A total of 41 sequences (6670 copies) belonging to the PiggyBac superfamily was identified in the *H. armigera* genome. Further analysis revealed that only four among the identified sequences showed similarity ranging from 81% to 99% with already identified PiggyBac transposons, npiggyBac-8 and piggyBac-2.

The following superfamilies of Class II elements, i.e., Transib, P elements, CACTA, Mutator, PIF-harbinger, Helitron, Maverick, Crypt-on and Merlin, were less represented in the *H. armigera* genome corresponding to a total of 7701 copies.

In addition, the MITE tracker allowed for the identification of 4185 putative MITEs corresponding to 43.11% of all annotated TEs. The analysis of terminal TIRs and TSD sequences allowed for the classification of 3570 MITE sequences (26,565 copies) into seven superfamilies (Table 2). The Tc1\mariner and PIF-Harbinger superfamilies were the most represented, with 1817 and 1368 MITEs, respectively, followed by the CACTA superfamily with 250 MITEs, then PiggyBac with 93 elements. The hAT, Transib and Maverick superfamilies were represented by only 20, 16 and six MITEs, respectively.

### 3.2. TE Insertions Scanning in Defensome Genes

Nucleotide BLAST searches for defensome genes in the regions framing identified TE sequences led to the identification of nine TE insertion sites in seven genes encoding for detoxifying enzymes (Table 3). The involved TEs are members of RTE, R2, CACTA, DIRS, Mariner, and hAT superfamilies (Table 3). Further analysis of TE insertions has shown that five TEs were inserted in four cytochrome P450 genes, an element was retrieved in a GST gene, two were hosted by ABC-G transporter gene and one was inserted in an ABC-C1 transporter gene.

Six of the inserted TEs have intronic insertion sites and one TE insertion occurred in the first exon of the ABCG transporter member 20 gene (Figure 4 and Appendix A). It should be noted that, for the cytochrome P450 4C1-like (LOC11300634) gene, harboring two LINE insertions, no exon or intron information was retrieved in GenBank.

## 4. Discussion

The present study identified TEs present in the genome of *H. armigera* and searched for their occurrence among the defensome genes of this pest. Characterizing TEs is an important task for non-model organisms and several TE annotation tools have recently been developed for TE characterization in these organisms [43]. In this work, we used the PiRATE pipeline [28], which combines different TE identification tools to detect, classify and annotate TEs into known superfamilies.

The results revealed a total TE content of 4336 sequences, covering 12.86% of the *H. armigera* assembly which is much higher than the previous data from Pearce et al. [29], in which Repeat Masker and Repeat Modeller tools allowed for the identification of a TE content representing only 0.88% of the genome. This confirms the interest of using a pipeline like PiRATE to increase the detection of TEs in a genome.

In other lepidopteran genomes, such as *Bombyx mori* and *Spodoptera frugiperda*, about 53% of TEs were previously identified as Class I and half of these belong to the *LINE* order [44]. Consistent with these results, 46.71% of the TEs identified in the *H. armigera* genome were Class I elements and the two most abundant were the *SINE* and *LINE* orders. These results suggest that using a combined approach is more specific for short TEs than the use of a single tool.

Some TEs were previously identified in *H. armigera* by molecular biology methods. In 2008, Sun et al. identified two *PiggyBac* elements (HaPLE1 and HaPLE2) in the cotton bollworm genome [45]. A few years later, multiple copies of two distinct mariner elements, *Hamar1* and *Hamar2*, were isolated by Wang et al. [46]. All these sequences were retrieved in the current study with a minimum of 88% identity.

To pinpoint TE insertions in defensome genes, the upstream and downstream regions of TE sequences were scanned and nine insertions have been successfully identified. Among them, five were retrieved in cytochrome P450 genes introns. Cytochrome P450 are among the three main groups of detoxification enzymes used by insects that play crucial roles in the detoxification of most pesticides [47]. In *H. armigera*, two *LINEs* (RTE and R2) and a *CACTA* element were inserted in the P450 4C1-like gene, the expression of which is responsible for the insecticide detoxification [48]. In *Aphis gossypii*, another LINE, the *Jockey* element was also identified in cytP450 6K1-like gene intron, putting it as the strongest candidate conferring resistance to thiamethoxam [49]. Two other *LINE* elements were inserted into the P450 4g15-like gene, but no information for exon or intron regions was retrieved in GenBank for this gene, suggesting that the *H. armigera* genome needs more annotation. However, the expression of this gene in *Aphis gossypii* was correlated with imidacloprid resistance [50].

Arthropod ABC transporters also play an important role in metabolite detoxification [10]. Three TEs, which belong to *RTE*, *mariner* and *hAT* superfamilies, were inserted in an ABCG transporter and the gene in the *H. armigera* genome. TE insertions into non-coding regions may be less subject to selection and could be influenced by other mechanisms of control. These insertions could be successfully spliced out during mRNA processing and thus may have no obvious effects on the function of the corresponding defensome gene [51]. Alternatively, they could result in exon skipping, alternative splicing, or alterations in expression profiles if the corresponding IR gene introns contain regulatory sequences, as exemplified by the insertion of the *Mu* element into an intron of the *Knotted* locus in Maize [51]. Analyses of the transcript and the expression levels of the identified IR genes as well as toxicity bioassay will be necessary to determine the exact effects of the nine identified TE insertions on the expression and function of the identified IR genes, as well as on the fitness of this species in the presence of a stressful environment such as insecticides.

## 5. Conclusions

Genome-wide TE annotation has rarely been performed in *H. armigera*. This study opens the way to new searches about the role of TEs in the genome evolution of *H. armigera* and their contribution to the pest adaptation such as insecticide resistance. In the present study, we conducted an accurate TE annotation, and the results reveal a total of 8521 TEs covering 12.86% (43349853 bp) of the *H. armigera* genome. The annotation of TEs was crossed with insertion sites search in defensome genes. Nine TEs belonging to the RTE, R2, LTR and TIR superfamilies were found to be inserted in CYP450, GST and ABC transporter genes and their insertion sites were mostly in intronic regions except for a hAT element inserted in the exon region. These results present for the first-time evidence that TEs are present in IR genes of *H. armigera*. However, further studies are necessary to elucidate the functional relationship of TEs with IR genes in *H. armigera*.

## Figures and Tables

**Figure 1 insects-11-00879-f001:**
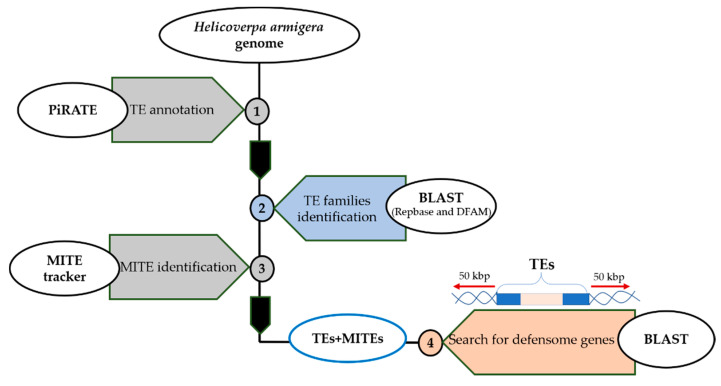
Flowchart for transposable elements annotation and insertion sites identification in the *H. armigera* genome.

**Figure 2 insects-11-00879-f002:**
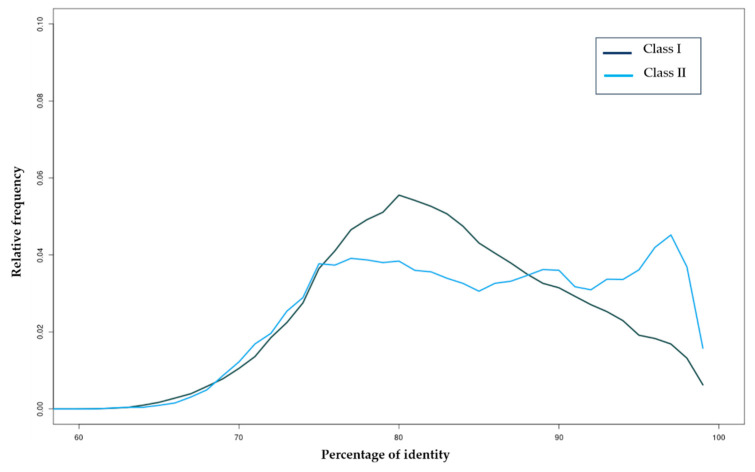
Distribution of sequence identity values between TE copies and TE sequences with at least one full-length copy for Class I and Class II elements. The relative frequencies per percentage of identity of Class I and Class II are represented in different colors.

**Figure 3 insects-11-00879-f003:**
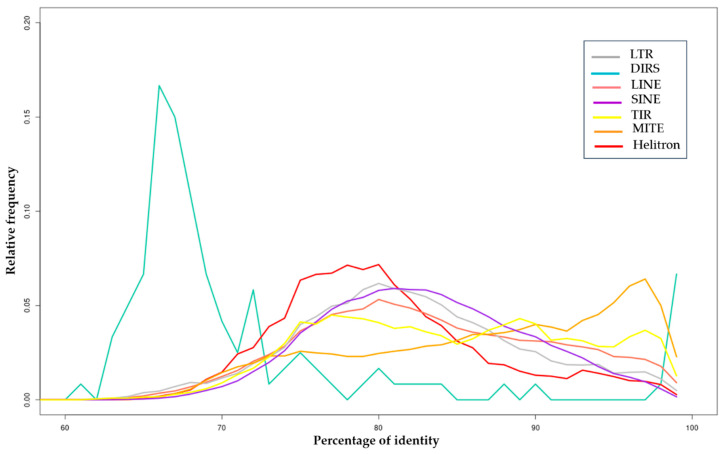
Distribution of sequence identity values between TE copies and TE sequences with at least one FLC. The relative frequencies per percentage of identity of Dictyostelium transposable element (DIRS), Helitron, Long and Short Interspersed Nuclear Elements (LINE and SINE), Long Terminal Repeat (LTR), Miniature Inverted Transposable Element (MITE) and Terminal Inverted Repeat (TIR) orders are represented in different colors. Only the main orders (in terms of copy number) are represented.

**Figure 4 insects-11-00879-f004:**
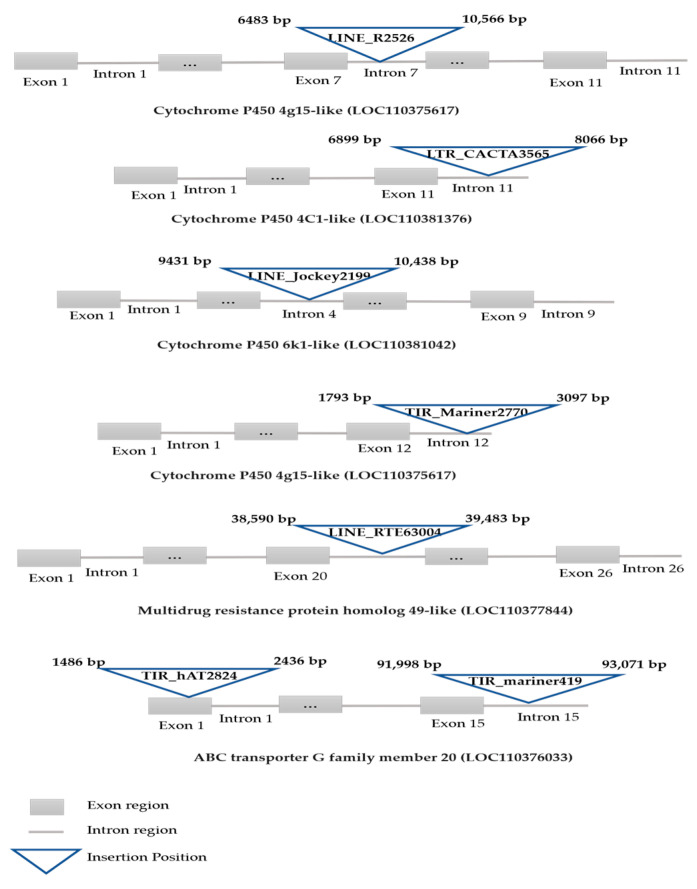
Schematic representation of TE insertion sites in genes encoding for detoxifying enzymes in *H. armigera*.

**Table 1 insects-11-00879-t001:** Summary of the identified and annotated TEs in the *H. armigera* genome.

Class	Order	Superfamily	Total Sequences ^1^	Full-Length Copies ^2^	TE Percentage	Copy Number ^3^
**Class I**	**LTR**	**Gypsy**	241	181		14,876
		**Bel pao**	155	97		1792
		**Copia**	77	67		223
	**Total LTR**		473	345	5.55%	16,891
	**DIRS**	**DIRS**	45	38		149
	**Total DIRS**		45	38	0.53%	149
	**LINE**	**Jockey**	716	480		44,432
		**RTE**	604	543		46,458
		**I**	172	93		16,520
		**R2**	169	101		10,614
	**Total LINE**		1661	1217	19.49%	118,024
	**SINE**	**tRNA**	1797	1120		51,581
		**5S**	4	-		0
	**Total SINE**		1801	1120	21.13%	51,581
**Total Class I**			3980	2720	46.71%	186,645
**Class II Subclass I**	**TIR**	**hAT**	120	105		6267
		**Mariner**	106	74		2284
		**Piggybac**	41	32		6670
		**Transib**	40	32		188
		**P**	15	12		6103
		**CACTA**	14	11		201
		**Mutator**	6	4		477
		**PIF_harbinger**	6	5		90
		**Merlin**	1	1		30
	**Total TIR**		349	276	4.09%	22,310
**Class II Subclass II**	**Crypton**	**Crypton**	1	1		3
	**Helitron**	**Helitron**	4	4		595
	**Maverick**	**Maverick**	2	2		14
	**MITEs**	**MITEs**	4185	130	49.11%	26,565
	**Total classII**		4541	413	53.29%	49,487
**Total TEs**			8521	3133	100	236,132

^1^ Representative sequence identified with PiRATE Step1 with an identity ≤ 80% ^2^ TEs sequences of PiRATE step 1 that align at least with one Full-length Copy (FLC) on the genome assembly ^3^ Copies annotated with TEannot REPET package v3.0. Number of undefined transposable elements are not shown in the table.

**Table 2 insects-11-00879-t002:** Distribution of MITEs identified in the *H. armigera* genome.

Superfamily	TSD	Number of MITEs	MITE Length (bp)	TIR Length (bp)
Tc1/mariner	TA	1817	50–360	10–21
PIF-Harbinger	TWA	1368	55–685	15–32
CACTA	2–3 bp	250	78–775	10–26
Piggybac	TTAA	93	50–800	15–31
hAT	TNNNNA	20	56–260	17–29
Transib	CNNNG	16	83–386	13–27
Maverick	6 bp	6	50–800	10–24
Other	-	615	50–800	11–35

**Table 3 insects-11-00879-t003:** TE insertions in genes encoding for detoxifying enzymes in *H. armigera.*

Gene Family	Gene Name	Gene Length (bp)	Inserted Element Name	TE Length (bp)	Insertion Position
Cytochrome P450 (CYP450)	4g15-like (LOC110375617)	14,974	LINE_R2_526	4083	6483–10,566
	4C1-like (LOC113006340)	39,707	LINE_RTE17512	620	5655–6274
			LINE_RTE84107	740	7021–7763
	4C1-like (LOC110381376)	8332	LTR_CACTA3565	1179	6899–8066
	6k1-like (LOC110381042)	15,958	LINE_jockey2199	1629	9431–10,438
Glutathione S-transferase (GST)	GST 1-like (LOC110371343)	4001	TIR_Mariner2770	1304	1793–3097
ATP binding cassette (ABC) transporter	ABC-G member 20 (LOC110376033)	96,146	TIR_Mariner419	1085	91,998–93,071
			TIR_hAT2824	951	1486–2436
	ABC-C1 (Multidrug resistance protein homolog 49-like MRP1) (LOC110377844)	48,302	LINE_RTE63004	894	38,590–39,483

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
