# Peer review of "Screening of Helicoverpa armigera Mobilome Revealed Transposable Element Insertions in Insecticide Resistance Genes"

_insects, 2020, doi:10.3390/insects11120879_

Round 1

Reviewer 1 Report

The Manuscript ID: insects-1000438 “ Screening of Helicoverpa armigera Mobilome Revealed Transposable Elements Insertions in Insecticide Resistance Genes” describe the transposable elements found in the H. armigera genome, which were characterized using the PiRATE pipeline, complemented by a seach for MITEs using MITE tracker. The mobilome was well characterized, and the results are interesting, being a significant contribution to the knowledge of the genome of this important agricultural pest. In addition, some TE insertions in insecticide resistance genes were characterized. This information can be useful for future searches involving TE and insecticide resistance in this species. I think the MS can be accepted for publication in INSECT journal.

Some minor points that I suggest can be addressed by the authors aiming improve the MS.

1) The authors could highlight the MITEs is not a “group” of TEs (as order, superfamily, family…) but a structural classification of some class II TEs. In the Table 2, is showed that the MITEs can be classified in their specific families. However, in the text it is not clear, and in some parts can promote confusion to readers.

2) The characterization of TEs families members in full length and partial (degenerated) copies is very interesting. However, I think a TE landscapes figure could help to better show these data.

TE landscapes are graphical representations of the proportion of each TE superfamily/family plot against the genetic distance observed between the TE copies and the consensus (ancestral element) derived from these copies, as a proxy for the TE superfamily/family age. These graphics allow an easy perception of the overall TE diversity and dynamic found in a given genome. TE landscapes can be built using the RepeatMasker on- line tool (http://www.repeatmasker. org/genomicDatasets/RMGenomicDatasets.html).

3) The TEs sequences found in the Helicoverpa armigera genome should made available, mainly the full length elements, in the supplementary material and/or in a TE database.

Author Response

Reviewer #1

The Manuscript ID: insects-1000438 “ Screening of Helicoverpa armigera Mobilome Revealed Transposable Elements Insertions in Insecticide Resistance Genes” describe the transposable elements found in the H. armigera genome, which were characterized using the PiRATE pipeline, complemented by a seach for MITEs using MITE tracker. The mobilome was well characterized, and the results are interesting, being a significant contribution to the knowledge of the genome of this important agricultural pest. In addition, some TE insertions in insecticide resistance genes were characterized. This information can be useful for future searches involving TE and insecticide resistance in this species. I think the MS can be accepted for publication in INSECT journal.

Answer: thank you for your positive feedback.

Some minor points that I suggest can be addressed by the authors aiming improve the MS.

  • The authors could highlight the MITEs is not a “group” of TEs (as order, superfamily, family…) but a structural classification of some class II TEs. In the Table 2, is showed that the MITEs can be classified in their specific families. However, in the text it is not clear, and in some parts can promote confusion to readers.

Answer: Following reviewer’s recommendation a brief description of MITEs is provided at lines 90-93, which may avoid any confusion.

2) The characterization of TEs families members in full length and partial (degenerated) copies is very interesting. However, I think a TE landscapes figure could help to better show these data.

 TE landscapes are graphical representations of the proportion of each TE superfamily/family plot against the genetic distance observed between the TE copies and the consensus (ancestral element) derived from these copies, as a proxy for the TE superfamily/family age. These graphics allow an easy perception of the overall TE diversity and dynamic found in a given genome. TE landscapes can be built using the RepeatMasker on- line tool (http://www.repeatmasker. org/genomicDatasets/RMGenomicDatasets.html).

Answer:  work is in progress we are at the final step for generating the TE landscape.  We will send you the final version of the manuscript with the TE landscape figure by the 10th of December.

3) The TEs sequences found in the Helicoverpa armigera genome should made available, mainly the full length elements, in the supplementary material and/or in a TE database.

Answer: the sequences of full length TEs identified in Helicoverpa armigera genome are now provided as File S1 and File S2, also in line 135.

Reviewer 2 Report

Review of Klai et al., Insects

General Comments

This is a simple but interesting bioinformatic analysis aimed to annotate the mobilome of H. armigera. The authors focus on TE inserted in insecticide resistance genes.

In particular, TEs Insertions were found in four genes encoding cytochrome P450, one encoding glutathione S-transferase, and two encoding ATP-binding cassette transporter.

Notably, these genes encode for detoxifying enzymes. There is no evidence (from literature or experiments within this paper) that they are involved in insecticide resistance.

It would be more interesting to investigate TE insertions in both resistant and susceptible strains, or in genes that have been actually linked to insecticide resistance (i.e. see ref 14, 15 and 16).

Mapping the TEs inserted in genes encoding for the detoxifying enzymes is only the first step. As the authors themself state “Analyses of the transcript and the expression levels of the identified IR genes as well as toxicity  bioassay will be necessary to determine the exact effects of the nine identified TEs insertions on the  expression and function of the identified IR genes, as well as on the fitness of this species in the  presence of stressful environment such as insecticides” (lines 237-240).

Anyway, I believe this is a valuable contribution to the field for future studies. 

Line 68: add references:

De Marco, et al.  Scien. Rep. 2017, 7, 41312

Dermauw, W.; van Leeuwen, Insect. Biochem. Mol. Biol. 2014, 45, 89–110

Steinberg, C. E. W. Arms Race Between Plants and Animals: Biotransformation System in Stress ecology: environmental stress as ecological driving force and key player in evolution (ed. Steinberg, C. E. W.) 61–105 (Springer New York, 2012).

Line 93:

the authors mapped the TEs inserted in genes encoding for the detoxifying enzymes (i.e. defensome, see De marco et al 2016 and references therein).

Change  “in genes conferring insecticide resistance” with “in the defensome” or “in the genes encoding for detoxifying enzymes”

See also line 185 (IR genes), line 199, lines 218, 249-250.

Author Response

Reviewer #2

This is a simple but interesting bioinformatic analysis aimed to annotate the mobilome of H. armigera. The authors focus on TE inserted in insecticide resistance genes.

In particular, TEs Insertions were found in four genes encoding cytochrome P450, one encoding glutathione S-transferase, and two encoding ATP-binding cassette transporter.

Notably, these genes encode for detoxifying enzymes. There is no evidence (from literature or experiments within this paper) that they are involved in insecticide resistance.

Bass, C., & Field, L. M. (2011). Gene amplification and insecticide resistance. Pest management science, 67(8), 886-890.

Papapostolou, K. M., Riga, M., Charamis, J., Skoufa, E., Souchlas, V., Ilias, A., ... & Vontas, J. (2020). Identification and characterization of striking multiple‐insecticide resistance in a Tetranychus urticae field population from Greece. Pest Management Science.

It would be more interesting to investigate TE insertions in both resistant and susceptible strains, or in genes that have been actually linked to insecticide resistance (i.e. see ref 14, 15 and 16).

Mapping the TEs inserted in genes encoding for the detoxifying enzymes is only the first step. As the authors themself state “Analyses of the transcript and the expression levels of the identified IR genes as well as toxicity  bioassay will be necessary to determine the exact effects of the nine identified TEs insertions on the  expression and function of the identified IR genes, as well as on the fitness of this species in the  presence of stressful environment such as insecticides” (lines 237-240).

Anyway, I believe this is a valuable contribution to the field for future studies.

Answer: Thank you for your positive feedback and encouragement. The present study was dedicated to screen the mobilome in the genome of Helicoverpa armigera with a focus of TE inserted in pesticide resistance genes. The inventory of TE linked to resistance genes is now completed. We think that the present work constitutes a necessary first step to target resistance genes containing TEs. Based on the present work, further molecular searches will assess their impact on gene expression. The functional study is out of the scope of the present work. However, it is going to be a very interesting goal in the future and will probably be the research topic of a future Ph D. student with a molecular approach. In addition, such functional studies would take a very long time with no certainty of results, which is far beyond a revision of the manuscript.

Line 68: add references:

De Marco, et al.  Scien. Rep. 2017, 7, 41312

Dermauw, W.; van Leeuwen, Insect. Biochem. Mol. Biol. 2014, 45, 89–110

Steinberg, C. E. W. Arms Race Between Plants and Animals: Biotransformation System in Stress ecology: environmental stress as ecological driving force and key player in evolution (ed. Steinberg, C. E. W.) 61–105 (Springer New York, 2012).

Answer: All these three references have been added accordingly at line 64 and 71, respectively

Line 93:

the authors mapped the TEs inserted in genes encoding for the detoxifying enzymes (i.e. defensome, see De marco et al 2016 and references therein).

Change  “in genes conferring insecticide resistance” with “in the defensome” or “in the genes encoding for detoxifying enzymes”

See also line 185 (IR genes), line 199, lines 218, 249-250.

Answer: The term IR was changed by “defensome” when the context was a general context, and by “in the genes encoding for detoxifying enzymes” when the context was a specific one. Line 97, 126, 128, 189-191, 200, 202, 206, 226, 254.

Reviewer 3 Report

In their manucript, authors show that in the cotton bollworm, Helicoverpa armigera, nine transposable element (TE) insertions are present in insecticide resistance (IR) genes encoding cytochrome P450 (CyP450), glutathione S-transferase (GST), and ATP-binding cassette (ABC) transporter. This is not novel since many studies have already shown that insecticide resistance can be associated with TEs insertions in specific genes.  The here presented evidence is solely based on H. armigera mobilome annotation and I fully agree with the authors that "further studies are necessary to elucidate a functional relationship between TEs and IR genes". However, they do not give such proof. 

In my opinion, the manuscript should be accepted only after adding some functional studies, which support such effects of TEs.

Minors: 

  • acronyms used in the Abstract have to be explained. The same is true for the Introduction
  • line 36: were less represented and accont...
  • line 120 and others, including tables and figures: what is pb? Do authors mean bp?
  • line 189: and one was inserted...
  • References are not written as shown in MDPI Author's Instructions

Author Response

Reviewer #3

In their manuscript, authors show that in the cotton bollworm, Helicoverpa armigera, nine transposable element (TE) insertions are present in insecticide resistance (IR) genes encoding cytochrome P450 (CyP450), glutathione S-transferase (GST), and ATP-binding cassette (ABC) transporter. This is not novel since many studies have already shown that insecticide resistance can be associated with TEs insertions in specific genes.  The here presented evidence is solely based on H. armigera mobilome annotation and I fully agree with the authors that "further studies are necessary to elucidate a functional relationship between TEs and IR genes". However, they do not give such proof.

In my opinion, the manuscript should be accepted only after adding some functional studies, which support such effects of TEs.

Answer: The present study was dedicated to screen the mobilome in the genome of Helicoverpa armigera with a focus of TE inserted in pesticide resistance genes. The inventory of TE linked to resistance genes is now completed. We think that the present work constitutes a necessary first step to target resistance genes containing TEs. Based on the present work, further molecular searches will assess their impact on gene expression. The functional study is out of the scope of the present work. However, it is going to be a very interesting goal in the future and will probably be the research topic of a future Ph D. student with a molecular approach. In addition, such functional studies would take a very long time with no certainty of results, which is far beyond a revision of the manuscript.

Minors:

acronyms used in the Abstract have to be explained. The same is true for the Introduction

We have explained the acronyms in the abstract and introduction.

line 36: were less represented and accont...

This sentence has been modified.

line 120 and others, including tables and figures: what is pb? Do authors mean bp?

Sorry for this mistake, yes it was bp and this has been changed.

line 189: and one was inserted...

This has been changed (line 194)

References are not written as shown in MDPI Author's Instructions

This has been corrected accordingly.

Round 2

Reviewer 3 Report

I do not agree with Authors' statement that functional studies should be out of scope of the present paper. It is clear that such studies would take some time. However, the question of time should not stand in the way of scientific progress.

There are still many misspellings in the References (species names in italics; second part of a species names in lowercase letters, e.g. Drosophila melanogaster) etc.

  • line 38: species name in italics

Author Response

Reviewer #3

I do not agree with Authors' statement that functional studies should be out of scope of the present paper. It is clear that such studies would take some time. However, the question of time should not stand in the way of scientific progress.

There are still many misspellings in the References (species names in italics; second part of a species names in lowercase letters, e.g. Drosophila melanogaster) etc.

line 38: species name in italics

Answer : We are thankful for the minor comments which have all been addressed to improve the manuscript and we are sorry to not  be able to undergo the functional work as we do not have the experimental expertise for such study.

Round 3

Reviewer 3 Report

(1) Corrections in Fig. 4 have not been done (bp instead of pb in all cases)

(2) References are still not correct: species names (e.g. Glyphodes pyloalis and not Glyphodes Pyloalis); use of lowercase and uppercase letters in Ref. 45; species names in italics in Ref. 46, 47

Author Response

Dear Reviewer,

We have done all suggested corrections and we hope that this revised version meets insects’s publication criteria.

Thank you again for your careful review.

Pr Maha MEZGHANI KHEMAKHEM
